# VARIATIONAL SPARSE CODING

## ABSTRACT

Variational auto-encoders (VAEs) offer a tractable approach when performing approximate inference in otherwise intractable generative models. However, standard VAEs often produce latent codes that are disperse and lack interpretability, thus making the resulting representations unsuitable for auxiliary tasks (e.g. classification) and human interpretation. We address these issues by merging ideas from variational auto-encoders and sparse coding, and propose to explicitly model sparsity in the latent space of a VAE with a Spike and Slab prior distribution. We derive the evidence lower bound using a discrete mixture recognition function thereby making approximate posterior inference as computational efficient as in the standard VAE case. With the new approach, we are able to infer truly sparse representations with generally intractable non-linear probabilistic models. We show that these sparse representations are advantageous over standard VAE representations on two benchmark classification tasks (MNIST and Fashion-MNIST) by demonstrating improved classification accuracy and significantly increased robustness to the number of latent dimensions. Furthermore, we demonstrate qualitatively that the sparse elements capture subjectively understandable sources of variation.

## 1    INTRODUCTION

Variational auto-encoders (VAEs) offer an efficient way of performing approximate posterior inference with otherwise intractable generative models and yield probabilistic encoding functions that can map complicated high-dimensional data to lower dimensional representations (Kingma & Welling, 2013; Rezende et al., 2014; Sønderby et al., 2016; Rasmus et al., 2015). Making such representations meaningful and efficient, however, is a particularly difficult task and currently a major challenge in representation learning (Hsu et al., 2017; Burgess et al., 2018; Kim & Mnih, 2018; Tomczak & Welling, 2017). Large latent spaces often give rise to many latent dimensions that do not carry any information, and obtaining codes that properly capture the complexity of the observed data is generally problematic (Tomczak & Welling, 2017; Higgins et al., 2016; Burgess et al., 2018).

In the case of linear mappings, sparse coding offers an elegant solution to the aforementioned problem; the representation space is induced to be sparse. In such a way, the encoding function is encouraged to use the minimum number of non-zero elements necessary to describe each observation and condense information in few active variables, different for each sample (Olshausen & Field, 1996a;b). In fact, due to their efficiency of representation, sparse codes have been used in many learning and recognition systems, as they provide easier interpretation (Lee et al., 2007; Bengio et al., 2013; Mairal et al., 2009; Arora et al., 2015) and increased efficiency in, for example, classification, clustering, and transmission tasks when used as learning inputs (Yang et al., 2011; Wright et al., 2009; Labusch et al., 2008; Yang et al., 2009).

In this work, we aim to extent the aforementioned capability of linear sparse coding to non-linear probabilistic generative models thus allowing efficient, informative and interpretable representations in the general case. To this end we formulate a new variation of the classical VAE in which we employ a sparsity inducing prior in the latent space based on the Spike and Slab distribution. We match this by a discrete mixture recognition function that can map observations to sparse latent vectors. Efficient inference, comparable in complexity to that of standard VAEs, is achieved by deriving an evidence lower bound (ELBO) for the new model which is optimized using standard gradient methods to recover the encoding and decoding functions. In our experiments, we consider two benchmark dataset (MNIST and Fashion-MNIST) and show how the resulting ELBO is able

to recover sparse, informative and interpretable representations regardless of the predefined number of latent dimensions. The ability to adjust to data complexity allows to automatically discover the sources of variation in given observations, without the need to carefully adjust the architecture of a model to the given representation task. We demonstrate these properties by first performing classification experiments using latent vectors as inputs, where we demonstrate that VSC representations marginally outperform VAE ones and display greatly improved robustness over large variations in latent space dimensionality. Secondly we show that many sparse elements in retrieved codes control subjectively recognisable features in the generated observations.

## 2 BACKGROUND AND RELATED WORK

### 2.1 SPARSE CODING

Sparse coding aims to approximately represent input vectors $x_i$ with a weighted linear combination of few unknown basis vectors $b_j$ (Lee et al., 2007; Bengio et al., 2013; Li et al., 2004). The problem of determining the optimal basis and weights is generally formulated as the minimisation of an objective function of the following form

$$\underset{B,Z}{\arg\min} \quad \frac{1}{2}||X - BZ||^2 + \lambda \sum_i \phi(z_i), \tag{1}$$

where $X \in \mathbb{R}^{M \times N}$ is the matrix of data, having as columns the input vectors $x_i \in \mathbb{R}^{M \times 1}$, $B \in \mathbb{R}^{M \times J}$ is the matrix having as columns the basis vectors $b_j \in \mathbb{R}^{M \times 1}$, $Z \in \mathbb{R}^{J \times N}$ is the sparse codes matrix, having as columns the sparse codes $z_i \in \mathbb{R}^{J \times 1}$ corresponding to the inputs $x_i$, $\lambda$ is a real positive parameter and $\phi(z_i)$ is a sparsity inducing function.

Sparse coding can be probabilistically interpreted as a generative model, where the observed vectors $x_i$ are generated from the unobserved latent variables $z_i$ through the linear process $x_i = Bz_i + \epsilon$, where $\epsilon$ is the observation noise and is drawn from an isotropic normal distribution with zero mean (Lee et al., 2007; Bengio et al., 2013). The model can then be described with the following prior and likelihood distributions

$$p(z_i) = \exp(-\beta\phi(z_i)), \qquad p(x_i|z_i) = \mathcal{N}(x_i; Bz_i, I\sigma^2), \tag{2}$$

where $\beta$ is a real positive parameter, $\sigma$ is the standard deviation of the observation noise and $I$ is the identity matrix. Performing maximum a posteriori (MAP) estimation with this model results in the minimisation shown in equation 1 with $\lambda = \sigma^2\beta$.

In contrast to the MAP formulation, we are interested in maximising the marginal likelihood $p(\mathbf{x}) = \prod p(x_i)$ and being able to perform such optimisation for arbitrarily complicated likelihood functions $p(x|z)$.

Previous work has demonstrated variational EM inference for such maximisation in the linear generative model case, with a particular choice of sparsity inducing prior (Titsias & Lázaro-Gredilla, 2011; Goodfellow et al., 2012). However, EM inference becomes intractable for more complicated non-linear posteriors and a large number of input vectors (Kingma & Welling, 2013), making such an approach unsuitable to scale to our desired model.

Conversely, some work has been done in generalising sparse coding to non-linear transformations, by defining sparsity on Riemannian manifolds (Ho et al., 2013; Cherian & Sra, 2017). These generalisations, however, perform MAP inference as they define a non-linear equivalent of the objective function in equation 1 and are limited to simple manifolds due to the need to compute the manifold's logarithmic map.

### 2.2 VARIATIONAL AUTO-ENCODERS

Variational auto-encoders (VAEs) are models for unsupervised efficient coding that aim to maximise the marginal likelihood $p(\mathbf{x}) = \prod p(x_i)$ with respect to some decoding parameters $\theta$ of the likelihood function $p_\theta(x|z)$ and encoding parameters $\phi$ of a recognition model $q_\phi(z|x)$ (Kingma & Welling (2013); Rezende et al. (2014); Pu et al. (2016)).

The VAE model is as follows; an observed vector $x_i \in \mathbb{R}^{M \times 1}$ is assumed to be drawn from a likelihood function $p_\theta(x|z)$. The likelihood function is chosen to fit the the expected nature of

Figure 1: Schematic representation of the variational sparse coding model (right) compared to a standard VAE (left). In both cases an observed variable $x_i$ is assumed to be generated from an unobserved variable $z_i$. Variational sparse coding, however, models sparsity in the latent space with the Spike and Slab prior distribution. One example is shown for each prior with a sample from the MNIST dataset.

variation in the observations. Common choices are a Gaussian or a Bernoulli distributions. The parameters of $p_\theta(x|z)$ are the output of a neural network having as input a latent variable $z_i \in \mathbb{R}^{J \times 1}$. The latent variable is assumed to be drawn from a prior $p(z)$ which can be chosen to take different parametric forms. In the most common VAE implementations, the prior takes the form of a multivariate Gaussian with identity covariance $\mathcal{N}(z; 0, I)$ (Kingma & Welling, 2013; Rezende et al., 2014; Higgins et al., 2016; Burgess et al., 2018; Yeung et al., 2017).

The aim is then to maximise a joint posterior distribution of the form $p(\mathbf{x}) = \prod \int p_\theta(x_i|z)p(z)dz$, which for an arbitrarily complicated conditional $p(x|z)$ is intractable. To address this intractability, VAEs introduce a recognition model $q_\phi(z|x)$ and define an evidence lower bound to be estimated in place of the true posterior.

The recognition function is a chosen to be a parametric distributions, where the parameters are the output of a neural network having as input a data point $x_i$. The ELBO can, due to Jensen's inequality, be formulated as

$$
\log p_\theta(x_i) = \log \int p_\theta(x_i|z)p(z)\frac{q_\phi(z|x_i)}{q_\phi(z|x_i)}dz \geq \mathcal{L}(\theta, \phi; x_i),
$$
$$
\mathcal{L}(\theta, \phi; x_i) = -D_{KL}(q_\phi(z|x_i)||p(z)) + \mathbb{E}_{q_\phi(z|x_i)}\left[\log p_\theta(x_i|z)\right].
$$
(3)

The ELBO is composed of two terms; a prior term, which encourages minimisation of the KL divergence between the encoding distributions and the prior, and a reconstruction term, which maximises the expectation of the data likelihood under the recognition function. The VLB is then maximised with respect to the model's parameters $\theta$ and $\phi$. The prior term can be defined analytically, while the reconstruction term is optimised stochastically through a reparameterization trick (Kingma & Welling, 2013). Figure 1 (left) schematically depicts the model with an example of data and corresponding latent variable.

## 2.3 DISCRETE LATENT VARIABLES AND SPARSITY IN VAES

Discrete latent distributions are a closely related theme to sparsity, as exactly sparse PDFs involve sampling from some discrete variables. Nalisnick & Smyth (2016) and Singh et al. (2017) model VAEs with a Stick-Breaking Process and an Indian Buffet Process priors respectively in order to allow for stochastic dimensionality in the latent space. In such a way, the prior can set to zero unused dimensions. However, the resulting representations are not truly sparse; the same elements are set to zero for every encoded observation. The scope of these works is dimensionality selection rather than sparsification.

Other models which present discrete variables in their latent space have been proposed in order to capture discrete features in natural observations. Rolfe (2016) model a discrete latent space composed of continuous variables conditioned on discrete ones in order to capture both discrete and continuous sources of variation in observations. Similarly motivated, van den Oord et al. (2017) perform variational inference with a learned discrete prior and recognition function. The resulting

latent spaces can present sparsity, depending on the choice of prior. However, they do not induce directly sparse statistics in the latent space.

Perhaps the most closely related work to our own is the Epitomic VAE (Yeung et al., 2017). In this work, the authors propose to learn a deterministic selection variable that dictates which latent dimensions the recognition function should exploit in the latent space. In such a way, different embeddings can exploit different combinations of variables, which achieves the goal of counteracting over-pruning. This approach does result into sparse latent variables. However, the method can be considered variational only in the continuous variables and the samples are not induced to present the statistics of a given sparse prior, but rather activate a constant number of elements in the latent vectors.

Differently from the aforementioned prior work, we aim to directly induce sparsity in a continuous latent space through a sparse PDF and find a suitable evidence lower bound to perform approximate variational inference.

## 3  VARIATIONAL SPARSE CODING

We propose to use the framework of VAEs to perform approximate variational inference with neural network sparse coding architectures. With this approach, we aim to discover and discern the non-linear features that constitute variability in data and represent them as few non-zero elements in sparse vectors.

We model sparsity in the latent space with a Spike and Slab probability density prior. The Spike and Slab PDF is a discrete mixture model which assigns point mass to null elements and therefore probabilistically models sparsity (Goodfellow et al., 2012; Titsias & Lázaro-Gredilla, 2011; Mitchell & Beauchamp, 1988). Because of this characteristic, this distribution has been used in various Bayesian sparse inference models (Seeger, 2008; Mohamed et al., 2011; Shelton et al., 2015; Hernández-Lobato et al., 2013).

The Spike and Slab distribution is defined over two variables; a binary spike variable $s_j$ and a continuous slab variable $z_j$ (Mitchell & Beauchamp, 1988). The spike variable is either one or zero with defined probabilities $\alpha$ and $(1 - \alpha)$ respectively and the slab variable has a distribution which is either a Gaussian or a Delta function centered at zero, conditioned on whether the spike variable is one or zero respectively. The prior probability density over the latent variable $z$ we are interested in is then

$$p_s(z) = \prod_{j=1}^{J} \left( \alpha \mathcal{N}(z_j; 0, 1) + (1 - \alpha)\delta(z_j) \right), \tag{4}$$

where $\delta(\cdot)$ indicates the Dirac delta function centered at zero. This choice of prior leads to the assumption that observed data is generated from sparse vectors in the latent space. The recognition function $q_\phi(z|x)$ is chosen to be a discrete mixture model of the form

$$q_\phi(z|x_i) = \prod_{j=1}^{J} \left( \gamma_{i,j} \mathcal{N}(z_{i,j}; \mu_{z,i,j}, \sigma_{z,i,j}^2) + (1 - \gamma_{i,j})\delta(z_{i,j}) \right), \tag{5}$$

where the distribution parameters $\mu_{z,i,j}$, $\sigma_{z,i,j}^2$ and $\gamma_{i,j}$ are the outputs of a neural network having parameters $\phi$ and input $x_i$. A description of the recognition function neural network can be found in appendix A.2. Similarly to the standard Spike and Slab distribution of equation 4, the distribution of equation 5 can be described with Spike variables, having probabilities of being one $\gamma_{i,j}$, and Slab variables having Gaussian distributions $\mathcal{N}(z_{i,j}; \mu_{z,i,j}, \sigma_{z,i,j}^2)$. On one side, this choice of recognition function allows for the posterior to match the prior, while on the other, the freedom to control the Gaussian moments and the Spike probabilities independently enables the model to encode information in the latent space. Figure 1 (right) schematically depicts the model with an example of an observation and corresponding latent sparse vector. A more detailed description of the model can be found in appendix A.

As in the standard VAE setting, we aim to perform approximate variational inference by maximising a lower bound. The ELBO we aim to maximise during training is of the form detailed in equation 3, with the Spike and Slab probability density function $p_s(z)$ of equation 4 as prior and the discrete

mixture distribution of of equation 5 as recognition function $q_\phi(z|x_i)$. In the following subsections we derive the prior and reconstruction terms of the VSC lower bound under these conditions.

## 3.1 ELBO PRIOR TERM

In this section we derive in closed form the regularisation component of the lower bound for our model, corresponding to the negative of the KL divergence between the discrete mixture of equation 5 and the Spike and Slab PDF.

As both $p_s(z_j)$ and $q_\phi(z_j|x)$ are mixtures of Dirac Delta functions and Gaussians, the regularisation term can be split in four cross entropy component in each latent dimension; two Gaussian-discrete mixture components and two Dirac Delta-discrete mixture components:

$$
\begin{aligned}
- D_{KL}(q_\phi(z|x_i)||p_s(z)) &= \int q_\phi(z|x_i)(\log p_s(z) - \log q_\phi(z|x_i))dz \\
&= \sum_j^J \Big[ \gamma_{i,j} \int \mathcal{N}(z_{i,j}; \mu_{z,i,j}, \sigma_{z,i,j}^2) \log\left[\alpha\mathcal{N}(z_j; 0, 1) + (1-\alpha)\delta(z_j)\right] dz_j \\
&\quad + (1-\gamma_{i,j}) \int \delta(z_{i,j}) \log\left[\alpha\mathcal{N}(z_j; 0, 1) + (1-\alpha)\delta(z_j)\right] dz_j \\
&\quad - \gamma_{i,j} \int \mathcal{N}(z_{i,j}; \mu_{z,i,j}, \sigma_{z,i,j}^2) \log\left[\gamma_{i,j}\mathcal{N}(z_{i,j}; \mu_{z,i,j}, \sigma_{z,i,j}^2) + (1-\gamma_{i,j})\delta(z_{i,j})\right] dz_j \\
&\quad - (1-\gamma_{i,j}) \int \delta(z_{i,j}) \log\left[\gamma_{i,j}\mathcal{N}(z_{i,j}; \mu_{z,i,j}, \sigma_{z,i,j}^2) + (1-\gamma_{i,j})\delta(z_{i,j})\right] dz_j \Big].
\end{aligned}
\tag{6}
$$

The first and third term have the form of a cross entropy between a Gaussian and a discrete mixture distribution. These components reduce to the corresponding weighted Gaussian-Gaussian entropy terms, as the point mass contributions vanish. In fact, for any finite density distributions $f(z_j)$ and $g(z_j)$, the point mass contribution to the cross entropy between $f(z_j)$ and a discrete mixture $h(z_j) = \alpha g(z_j) + (1-\alpha)\delta(z_j - c)$ is infinitesimal. The proof is as follows:

$$
\begin{aligned}
&\int f(z_j) \log\left[\alpha g(z_j) + (1-\alpha)\delta(z_j - c)\right] dz_j \\
&= \int \left(1 - \frac{\delta(z_j - c)}{\delta(0)}\right) f(z_j) \log\left[\alpha g(z_j)\right] + \\
&\quad + \frac{\delta(z_j - c)}{\delta(0)} f(z_j) \log\left[\alpha g(z_j) + (1-\alpha)\delta(z_j - c)\right] dz_j \\
&= \int f(z_j) \log\left[\alpha g(z_j)\right] dz_j + \lim_{u\to\infty} \frac{f(c)}{u} \log(1 + \frac{1-\alpha}{\alpha}\frac{u}{g(c)}).
\end{aligned}
\tag{7}
$$

Where the last term vanishes. Applying this result to the first and third cross entropy terms gives the corresponding standard weighted Gaussian-Gaussian result, plus the normalisation constant $\gamma_{i,j}\log(\alpha/\gamma_{i,j})$. The second and fourth terms take the form of the cross entropy between a Dirac Delta function and a discrete mixture distribution. In this case, instead, the continuous density contributions vanish:

$$
\begin{aligned}
&(1-\gamma_{i,j}) \int \delta(z_{i,j}) \big( \log\left[\alpha\mathcal{N}(z_j; 0, 1) + (1-\alpha)\delta(z_j)\right] \\
&= \lim_{u\to\infty} (1-\gamma_{i,j}) \log\left[\frac{\alpha\mathcal{N}(0; 0, 1) + (1-\alpha)u}{\gamma_{i,j}\mathcal{N}(0; \mu_{z,i,j}, \sigma_{z,i,j}^2) + (1-\gamma_{i,j})u}\right] \\
&= (1-\gamma_{i,j}) \log\left(\frac{1-\alpha}{1-\gamma_{i,j}}\right).
\end{aligned}
\tag{8}
$$

Combining the two results, we obtain the prior term of the VSC lower bound

$$
-D_{KL}(q_\phi(z|x_i)||p_s(z)) = \sum_j^J \left[ -\frac{\gamma_{i,j}}{2} \big(1 + \log(\sigma_{z,i,j}^2) - \mu_{z,i,j}^2 - \sigma_{z,i,j}^2\big) \right.
$$
$$
\left. + (1 - \gamma_{i,j}) \log\left(\frac{1-\alpha}{1-\gamma_{i,j}}\right) + \gamma_{i,j} \log\left(\frac{\alpha}{\gamma_{i,j}}\right) \right]. \tag{9}
$$

A more detailed derivation is provided in appendix B. This prior term naturally presents two components. The first is the negative KL divergence between the distributions of the Slab variables, multiplied by the probability of $z_{i,j}$ being non-zero $\gamma_{i,j}$. This component gives a similar regularisation to that of the standard VAE and encourages the Gaussian components of the recognition function to match those of the prior, proportionally to the Spike probabilities $\gamma_{i,j}$. The second term is the negative KL divergence between the distributions of the Spike variables. This term encourages the probabilities of the latent variables being non-zero $\gamma_{i,j}$ to match the prior Spike probability $\alpha$.

## 3.2 ELBO RECONSTRUCTION TERM

Similarly to the standard VAE, the reconstruction term of the lower bound is estimated and maximised stochastically as follows

$$
\mathbb{E}_{q_\phi(z|x_i)}\left[\log p_\theta(x_i|z)\right] \simeq \frac{1}{L}\sum_l^L \log p_\theta(x_i|z_{i,l}), \tag{10}
$$

where the samples $z_{i,l}$ are drawn from the recognition function $q_\phi(z|x_i)$. As in the standard VAE, to make the reconstruction term differentiable with respect to the encoding parameters $\phi$, we employ a reparameterization trick to draw from $q_\phi(z|x_i)$. To parametrise samples from the discrete binary component of $q_\phi(z|x_i)$ we use a continuous relaxation of binary variables analogous to that presented in Maddison et al. (2016) and (Rolfe, 2016). We make use of two auxiliary noise variables $\epsilon$ and $\eta$, normally and uniformly distributed respectively. $\epsilon$ is used to draw from the Slab distributions, resulting in a reparametrisation analogous to the standard VAE (Kingma & Welling, 2013). $\eta$ is used to parametrise draws of the Spike variables through a non-linear binary selection function $T(y_{i,l})$. The two variables are then multiplied together to obtain the parametrised draw from $q_\phi(z|x_i)$. A more detailed description of the reparametrisation of sparse samples is reported in appendix C

## 3.3 THE VSC EVIDENCE LOWER BOUND

Combining the prior and reconstruction terms from section 3.1 and 3.2, we obtain the estimation of the VSC lower bound

$$
\mathcal{L}(\theta, \phi; x_i) \simeq \sum_j^J \left[ \frac{\gamma_{i,j}}{2}\big(1 + \log(\sigma_{z,i,j}^2) - \mu_{z,i,j}^2 - \sigma_{z,i,j}^2\big) \right.
$$
$$
\left. + (1-\gamma_{i,j})\log\left(\frac{1-\alpha}{1-\gamma_{i,j}}\right) + \gamma_{i,j}\log\left(\frac{\alpha}{\gamma_{i,j}}\right) \right] + \frac{1}{L}\sum_l^L \log p_\theta(x_i|z_{i,l}). \tag{11}
$$

The final ELBO is relatively simple and of easy interpretation; the prior term is composed of the negative Spike and weighted Slab KL divergences, while the reconstruction term is the expectation of the likelihood under the recognition function PDF, estimated stochastically. We also point out that for $\gamma_{i,j} = \alpha = 1$ we recover the lower bound of the standard VAE (Kingma & Welling, 2013) as expected from the definition of the model. To train the VSC model, we maximise the ELBO in the form of equation 11 with respect to the encoding and decoding parameters $\phi$ and $\theta$ through gradient ascent.

## 4 EXPERIMENTS

We test the VSC model on two image datasets commonly used to benchmark learning performance; the hand written digits dataset MNIST (LeCun et al., 1998) and the more recent fashion items dataset

fashion-MNIST (Xiao et al., 2017), both composed of $28 \times 28$ grey scale images of handwritten digits and pieces of clothing respectively. We also make use of the CelebA faces dataset (Liu et al., 2015) to illustrate more qualitative results. Details of these datasets are given in appendix D.2. Various examples of latent sparse codes and corresponding reconstructions are shown in appendix E.1, while measurements of the latent space sparsity are presented in appendix E.2.

In the following subsections we test the VSC model in different settings. First, we evaluate the ELBO at varying prior sparsity and number of latent space dimensions. Secondly, to evaluate quantitatively representation efficiency in the latent space, we test classification using latent variables as inputs. Lastly, we qualitatively assess latent space interpretation by examining the effect of altering individual non-zero elements in the sparse codes. Details of the experimental conditions can be found in appendix D.

## 4.1 ELBO EVALUATION

We evaluate the ELBO at varying numbers of latent dimensions and different levels of prior sparsity $\alpha$. We first train a standard VAE at a varying number of latent dimensions imposing a limit of $20,000$ iterations. For each dimensionality, we find the best performing initial step size for the Adam optimiser (Kingma & Ba, 2014). We then use identical settings in each condition to test VSCs lower bound with different prior sparsity. Our evaluation performed on the test sets is shown in figure 2. Results for different iteration limits are included in appendix E.3.

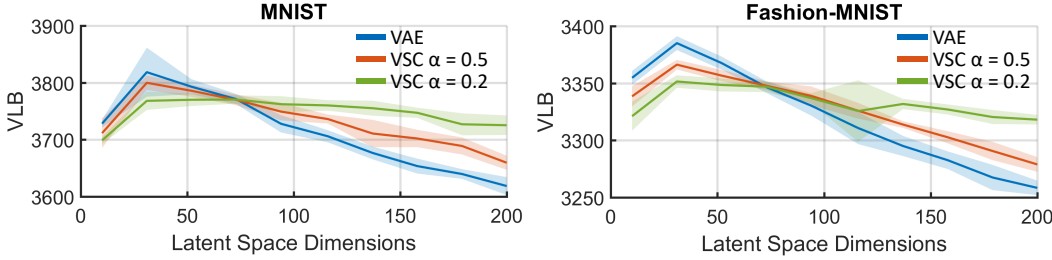

Figure 2: Test set ELBO for the VSC at varying number of latent dimensions. The standard VAE reaches high ELBO for a correct choice of latent space dimensions, but drops rapidly for larger latent spaces. With increasing sparsity in the latent space, the VSC drops in performance at the optimal VAE dimensionality, but remains more stable with larger latent spaces.

The standard VAE achieves high ELBO values provided that the size of its latent space is chosen correctly, but for spaces which are too large its performance rapidly drops, as encoding in many latent variables becomes increasingly difficult. Conversely, the VSC reaches a lower maximum ELBO, but remain significantly more stable with more latent dimensions. With few latent dimensions available, the sparsity imposed by the prior, controlled by the parameter $\alpha$, is too restrictive to allow rich descriptions of the observations and matching of the prior simultaneously. In this limit the ELBO of the VSC is comparable to that of a VAE, but slightly under-performs it. With more latent dimensions, only a subset of the available elements is used to encode each observation, making learning efficiency more stable as the latent space grows in size.

## 4.2 LEARNING IN THE LATENT SPACE

An important focus of this work is the ability of VSC to recover latent codes which carry a high level of information about the input. To test this aspect, we compare the representation efficiency of VAE and VSC by performing a standard classification experiment using the latent variables as input features. In order to encourage information rich codes in the VSC, we set the prior Spike probability $\alpha$ to a low value of $0.01$. With this very sparse prior, the recognition function activates non-zero elements only when needed to reconstruct an observation, while the prior induces the remaining elements to be mostly null.

We train VAEs and VSCs at varying number of latent dimensions for $20,000$ iterations. In each case, we use $5,000$ encoded labelled examples from the training sets to train a simple one layer fully

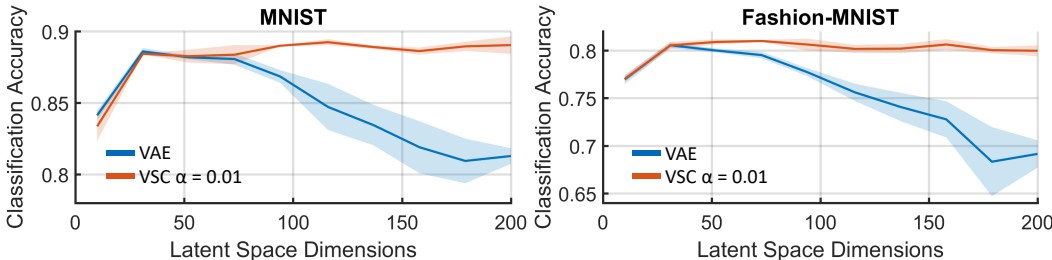

Figure 3: Classification performance of VSC and standard VAE at varying number of latent space dimensions. The VAE reaches its peak performance for optimal choice of latent space dimensions, but yields inefficient codes if the latent space is too large. VSC recovers efficient codes for arbitrarily large latent spaces which outperform the VAE ones as classification inputs.

connected classifiers using the latent codes as inputs. Figure 3 shows the classification performance obtained on the test set.

VSC is able to reliably recover efficient codes without the need to specify an optimal latent space size and also marginally outperforms the best VAE. This is because the recognition function activates only the subset of variables it needs to describe each observation, regardless of the latent space dimensionality. In such a way, the sources of variations in the observed data are automatically discovered and encoded into few non-zero elements. The peak performance for the VSCs occurs at larger latent spaces than for the standard VAEs, indicating that there is a representation advantage in encoding to larger spaces with sparser solutions than into smaller dense codes. Additional evaluations of classification accuracy at varying sparsity and number of labelled examples are shown in appendix E.4.

### 4.3 INTERPRETATION OF THE SPARSE CODES

Lastly, we qualitatively examine the interpretation of the non-zero elements in the sparse codes recovered with the VSC model. To this end, we encode several examples from the test sets of the Fashion-MNIST and CelebA datasets with VSCs trained with prior spike probability $\alpha = 0.01$. The Fashion-MNIST and CelebA examples were encoded in $200$ and $800$ latent dimensions respectively. We then show the effect of altering individual non-zero components on the reconstructed observations. Examples are shown in figure 4.

We find that many of the non-zero elements in the latent codes control interpretable features of the generated observations, as shown in figure 4. We further note that these results are not obtained through interpolation of many labelled examples, but simply by altering individually some of the few components activated by the recognition function. Though we are not directly inducing interpretation in the latent space, sparsity does lead to a higher expectation of interpretability due to the conditional activation of only certain dimensions. For a particular observation, the recognition function defines a low dimensional sub-space by activating only few non-zero elements that control the features necessary to describe such observation and similar ones, thereby defining a sort of sub-generative model for this type of objects (see appendix E.5 for examples of sampling in the sub-spaces defined by sparse encodings). For different observations, the model can activate different subsets of non-zero elements, exploiting a larger space for the aggregate posterior. In such a way, a particular example is described by a small subset of variables which are easier to manually explore, while the model can adjust its capacity to represent large and varied datasets.

It is also interesting to consider interpolation between different objects in the VSC latent space; as representations are sparse, so are interpolation vectors between them and we can examine their non-zero elements individually. We show an example considering the interpolation between one image of a shirt and one of a t-shirt in the Fashion-MNIST dataset. Figure 5 shows the effect of altering individually the two largest interpolation vector elements for each example.

The first and largest of the two non-zero elements considered controls the sleeves, which can be added to the t-shirt and subtracted from the shirt by altering this element alone. The second ele-

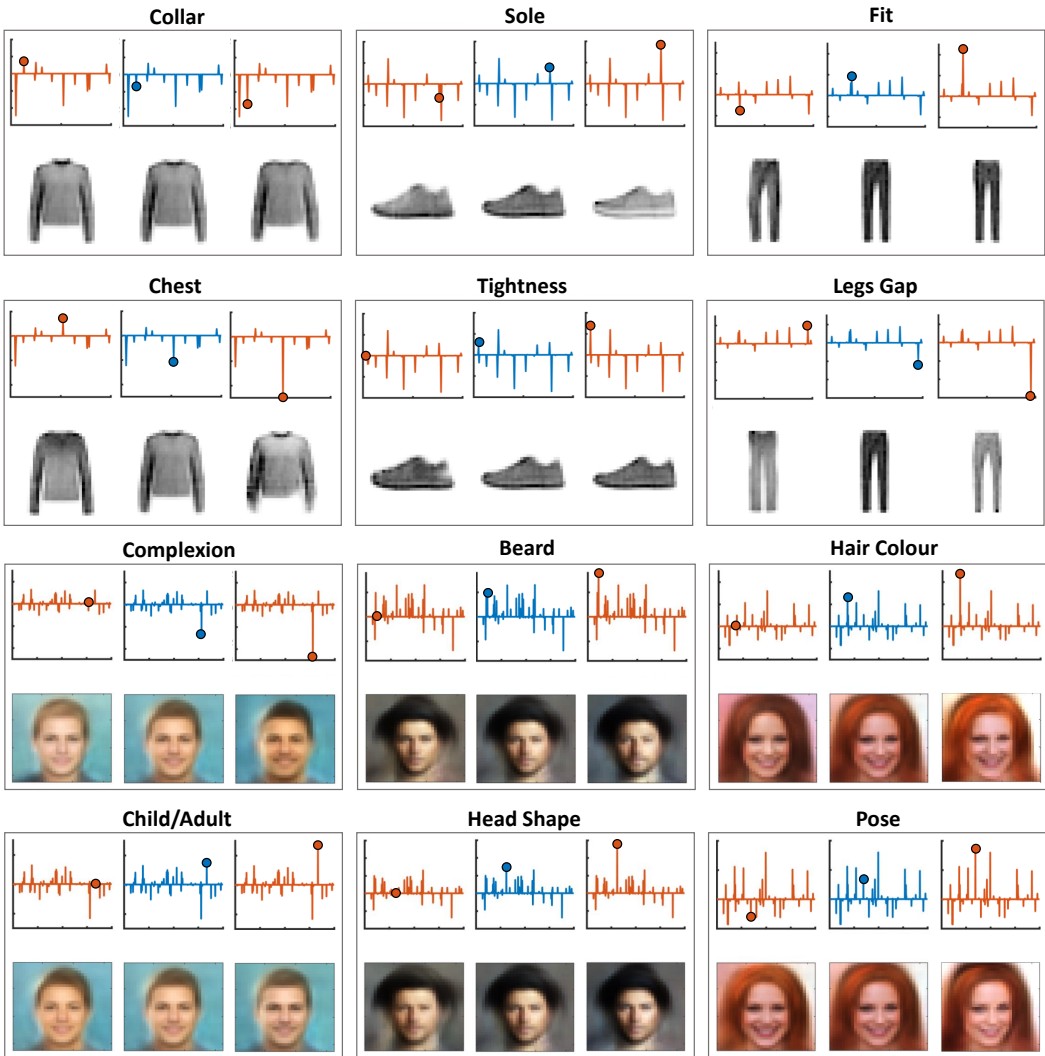

Figure 4: Effect on generation of altering single non-zero elements in VSC latent codes for Fashion-MNIST (Top) and CelebA (Bottom). The original latent codes are shown in blue and those altered in the latent space are shown in orange. Underneath each code the corresponding reconstruction is shown. The altered elements are highlighted with a coloured circle.

ment similarly controls the collar and buttons. Much like the non-zero elements of single encoded examples, those of the interpolation vector between similar objects seem to offer good interpretation.

## 5 CONCLUSION AND FUTURE WORK

In this paper, we lay the general framework to induce sparsity in the latent space of VAEs, allowing approximate variational inference with arbitrarily complicated and probabilistic sparse coding models. We derived a lower bound which is of clear interpretation and efficient to estimate and optimise, as the ELBO of a standard VAE. With the resulting encoders, we recovered efficient sparse codes, which proved to be optimal learning inputs in standard classification benchmarks and exhibit good interpretation in many of their non-zero components. We conclude that inducing sparsity in the latent space of generative models appears to be a promising route to obtaining useful codes, interpretable representations and controlled data synthesis, which are all outstanding challenges in VAEs and representation learning in general.

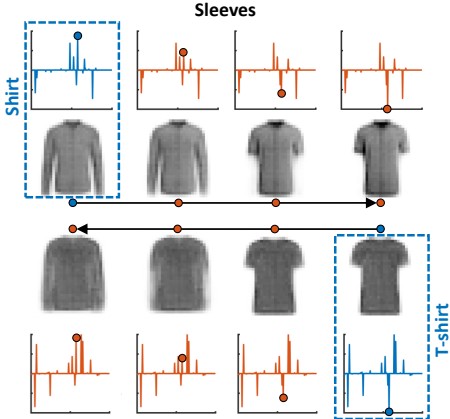 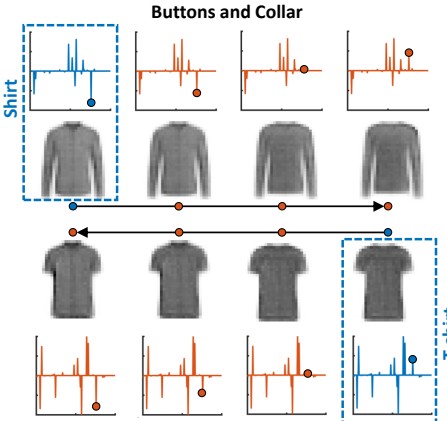

Figure 5: Effect of altering individual non-zero components of the interpolation vector between the sparse codes of two objects. The original latent codes are shown in blue and those altered in the latent space are shown in orange. The altered elements are highlighted with a coloured circle.

In future work, we aim to further study the properties of a sparse latent space with respect to its interpretation and features disentanglement capability. We expect VSC to be able to model huge ensembles of varied data by sparsely populating large latent spaces, hence isolating the features that govern variability among similar objects in widely diverse aggregates of data.

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

## A    DETAILS OF THE VSC MODEL

We describe here the details of the VSC model and the architecture of the neural networks we employed as likelihood and recognition functions.

### A.1    LIKELIHOOD FUNCTION

The likelihood function $p(x|z_i)$ is composed of a neural network which takes as input a latent variable $z_i \in \mathbb{R}^{J \times 1}$ and outputs the mean $\mu_i \in \mathbb{R}^{M \times 1}$ and log variance $\log(\sigma_i^2) \in \mathbb{R}^{M \times 1}$. The log likelihood of a sample $x_i$ is then computed evaluating the log probability density assigned to $x_i$ by a Gaussian having mean $\mu_i$ and standard deviation $\sigma_i$.

In our experiments we use a one hidden layer fully connected neural network for the VSCs trained with the MNIST and Fashion-MNIST datasets and a two hidden layers network for the VSCs trained with the CelebA dataset.

### A.2    RECOGNITION FUNCTION

The recognition function $p(z|x_i)$ is composed of a neural network which takes as input an observation $x_i \in \mathbb{R}^{M \times 1}$ and outputs the mean $\mu_{z,i} \in \mathbb{R}^{J \times 1}$, the log variance $\log(\sigma_{z,i}^2) \in \mathbb{R}^{J \times 1}$ and the log Spike probabilities vector $\log(\gamma_i) \in \mathbb{R}^{J \times 1}$. The elements of $\gamma_i$ need to be constrained between 0 and 1, therefore, other than using $\log(\gamma_i)$ as output, which ensures $\gamma_i > 0$, we employ a ReLU non-linearity at this output of the neural network as follows

$$\log(\gamma_i) = -ReLU(-v_{out,i}) \tag{12}$$

Where $v_{out,i}$ is output to the same standard neural network that outputs $\mu_{z,i}$ and $\log(\sigma_{z,i}^2)$. This ensures that $\gamma_i < 1$. Samples in the latent space $z_{i,l}$ can then be drawn as detailed in equation 24. As for the likelihood function, we use a one hidden layer fully connected neural network for the VSCs trained with the MNIST and Fashion-MNIST datasets and a two hidden layers network for the VSCs trained with the CelebA dataset.

## B    DERIVATION OF THE ELBO PRIOR TERM

We report here a detailed derivation of the VSC lower bound prior term shown in equation 9. As described in section 3, the lower bound we aim to maximise has the same form as the standard VAE one of equation 3, with the Spike and Slab probability density function $p_s(z)$ of equation 4 as prior and the discrete mixture distribution of of equation 5 as recognition function $q_\phi(z|x_i)$. The VSC

lower bound prior term is therefore obtained by substituting these distribution in the negative KL divergence term of equation 3. By doing so, we obtain four cross entropy components in each latent dimension

$$- D_{KL}(q_\phi(z|x_i)||p_s(z)) = \int q_\phi(z|x_i)(\log p_s(z) - \log q_\phi(z|x_i))dz$$

$$= \sum_j^J \Bigg[ \underbrace{\gamma_{i,j} \int \mathcal{N}(z_{i,j}; \mu_{z,i,j}, \sigma_{z,i,j}^2) \log\left[\alpha\mathcal{N}(z_j; 0, 1) + (1-\alpha)\delta(z_j)\right] dz_j}_{\text{\textcircled{1}}}$$

$$+ \underbrace{(1-\gamma_{i,j}) \int \delta(z_{i,j}) \log\left[\alpha\mathcal{N}(z_j; 0, 1) + (1-\alpha)\delta(z_j)\right] dz_j}_{\text{\textcircled{2}}} \tag{13}$$

$$- \underbrace{\gamma_{i,j} \int \mathcal{N}(z_{i,j}; \mu_{z,i,j}, \sigma_{z,i,j}^2) \log\left[\gamma_{i,j}\mathcal{N}(z_{i,j}; \mu_{z,i,j}, \sigma_{z,i,j}^2) + (1-\gamma_{i,j})\delta(z_{i,j})\right] dz_j}_{\text{\textcircled{3}}}$$

$$- \underbrace{(1-\gamma_{i,j}) \int \delta(z_{i,j}) \log\left[\gamma_{i,j}\mathcal{N}(z_{i,j}; \mu_{z,i,j}, \sigma_{z,i,j}^2) + (1-\gamma_{i,j})\delta(z_{i,j})\right] dz_j}_{\text{\textcircled{4}}} \Bigg].$$

①  and ③  are of a similar form; the cross entropy between a Gaussian and a discrete mixture distributions. These components reduce to the corresponding Gaussian-Gaussian entropy terms, as the point mass contributions vanish. In fact, for any finite density distributions $f(z_j)$ and $g(z_j)$, the point mass contribution to the cross entropy between $f(z_j)$ and a discrete mixture $h(z_j) = \alpha g(z_j) + (1-\alpha)\delta(z_j - c)$ is infinitesimal. The proof is as follows: The cross entropy between the functions $f(z_j)$ and $h(z_j)$ is

$$\int f(z_j) \log\left[\alpha g(z_j) + (1-\alpha)\delta(z_j - c)\right] dz_j. \tag{14}$$

We can split this integral in two components over two different domains, the first in the region where $z_j \neq c$ and the second in the region where $z_j = c$. By using a Dirac Delta function, the first component can be expressed as follows

$$\int_{z_j \neq c} f(z_j) \log\left[\alpha g(z_j) + (1-\alpha)\delta(z_j - c)\right] dz_j =$$
$$\int_{z_j \neq c} f(z_j) \log\left[\alpha g(z_j)\right] dz_j = \tag{15}$$
$$\int \left(1 - \frac{\delta(z_j - c)}{\delta(0)}\right) f(z_j) \log\left[\alpha g(z_j)\right],$$

where from the first to the second line we can ignore the component containing $\delta(z_j - c)$, as the domain does not include $z_j = c$. We then use a coefficient which is zero at $z_j = c$ and one otherwise to write the integral over the whole domain of $z_j$. Similarly, we can write the term in the domain $z_j = c$ as

$$\int_{z_j = c} f(z_j) \log\left[\alpha g(z_j) + (1-\alpha)\delta(z_j - c)\right] dz_j =$$
$$\int \frac{\delta(z_j - c)}{\delta(0)} f(z_j) \log\left[\alpha g(z_j) + (1-\alpha)\delta(z_j - c)\right] dz_j, \tag{16}$$

Now combining the two terms we obtain

$$\int f(z_j) \log \left[ \alpha g(z_j) + (1 - \alpha)\delta(z_j - c) \right] dz_j$$

$$= \int \left[ \left( 1 - \frac{\delta(z_j - c)}{\delta(0)} \right) f(z_j) \log \left[ \alpha g(z_j) \right] + \right. \tag{17}$$

$$\left. + \frac{\delta(z_j - c)}{\delta(0)} f(z_j) \log \left[ \alpha g(z_j) + (1 - \alpha)\delta(z_j - c) \right] \right] dz_j$$

Rearranging to gather the terms in $\delta(z_j - c)/\delta(0)$ we get

$$\int f(z_j) \log \left[ \alpha g(z_j) \right] dz_j +$$

$$\int \frac{\delta(z_j - c)}{\delta(0)} \left[ f(z_j) \log \left[ \alpha g(z_j) + (1 - \alpha)\delta(z_j - c) \right] - f(z_j) \log \left[ \alpha g(z_j) \right] \right] dz_j \tag{18}$$

$$= \int f(z_j) \log \left[ \alpha g(z_j) \right] dz_j + \int \frac{\delta(z_j - c)}{\delta(0)} f(z_j) \log \left[ \frac{\alpha g(z_j) + (1 - \alpha)\delta(z_j - c)}{\alpha g(z_j)} \right] dz_j.$$

Simplifying the argument of the second logarithm and solving the second integral we get

$$\int f(z_j) \log \left[ \alpha g(z_j) + (1 - \alpha)\delta(z_j - c) \right] dz_j$$

$$= \int f(z_j) \log \left[ \alpha g(z_j) \right] dz_j + \lim_{u \to \infty} \frac{f(c)}{u} \log(1 + \frac{1 - \alpha}{\alpha} \frac{u}{g(c)}), \tag{19}$$

where the second term tends to zero, leaving the cross entropy between $f(z_j)$ and $\alpha g(z_j)$. Applying this result to ①  and ③  we obtain the following

$$① - ③ = \gamma_{i,j} \int \left[ \mathcal{N}(z_{i,j}; \mu_{z,i,j}, \sigma_{z,i,j}^2) \log \left[ \alpha \mathcal{N}(z_j; 0, 1) \right] \right.$$

$$\left. - \mathcal{N}(z_{i,j}; \mu_{z,i,j}, \sigma_{z,i,j}^2) \log \left[ \gamma_{i,j} \mathcal{N}(z_{i,j}; \mu_{z,i,j}, \sigma_{z,i,j}^2) \right] \right] dz_j$$

$$= \gamma_{i,j} \int \mathcal{N}(z_{i,j}; \mu_{z,i,j}, \sigma_{z,i,j}^2) \log \left[ \frac{\alpha \mathcal{N}(z_j; 0, 1)}{\gamma_{i,j} \mathcal{N}(z_{i,j}; \mu_{z,i,j}, \sigma_{z,i,j}^2)} \right] dz_j \tag{20}$$

$$= -\gamma_{i,j} D_{KL}(\mathcal{N}(z_{i,j}; \mu_{z,i,j}, \sigma_{z,i,j}^2) \ || \ \mathcal{N}(z_j; 0, 1)) + \gamma_{i,j} \log \left( \frac{\alpha}{\gamma_{i,j}} \right)$$

The KL divergence $D_{KL} \left( \mathcal{N}(z_{i,j}; \mu_{z,i,j}, \sigma_{z,i,j}^2) \ || \ \mathcal{N}(z_j; 0, 1) \right)$ is analogous to that of the standard VAE and has a simple analytic form (Kingma & Welling, 2013):

$$D_{KL} \left( \mathcal{N} \left( z_{i,j}; \mu_{z,i,j}, \sigma_{z,i,j}^2 \right) || \ \mathcal{N} \left( z_j; 0, 1 \right) \right) = -\frac{1}{2} \left( 1 + \log \left( \sigma_{z,i,j}^2 \right) - \mu_{z,i,j}^2 - \sigma_{z,i,j}^2 \right) \tag{21}$$

②  and ④  take the form of the cross entropy between a Dirac Delta function and a discrete mixture distribution. In this case, instead, the continuous density contributions vanish:

$$\textcircled{2} - \textcircled{4} = (1 - \gamma_{i,j}) \int \delta(z_{i,j}) \big( \log \left[ \alpha \mathcal{N}(z_j; 0, 1) + (1 - \alpha) \delta(z_j) \right]$$
$$- \log \left[ \gamma_{i,j} \mathcal{N}(z_{i,j}; \mu_{z,i,j}, \sigma_{z,i,j}^2) + (1 - \gamma_{i,j}) \delta(z_{i,j}) \right] \big) dz_j$$
$$= \lim_{u \to \infty} (1 - \gamma_{i,j}) \log \left[ \frac{\alpha \mathcal{N}(0; 0, 1) + (1 - \alpha)u}{\gamma_{i,j} \mathcal{N}(0; \mu_{z,i,j}, \sigma_{z,i,j}^2) + (1 - \gamma_{i,j})u} \right] \qquad (22)$$
$$= (1 - \gamma_{i,j}) \log \left( \frac{1 - \alpha}{1 - \gamma_{i,j}} \right).$$

Substituting the results of equations 20, 21 and 22 into equation 13, we obtain the prior term of the VSC lower bound

$$-D_{KL}(q_\phi(z|x_i)||p_s(z)) = \sum_j^J \left[ \textcircled{1} - \textcircled{3} + \textcircled{2} - \textcircled{4} \right]$$
$$= \sum_j^J \left[ \gamma_{i,j} \underbrace{\frac{1}{2} \left( 1 + \log(\sigma_{z,i,j}^2) - \mu_{z,i,j}^2 - \sigma_{z,i,j}^2 \right)}_{\text{Negative Slab KL Divergence}} \right.$$
$$\left. + \underbrace{(1 - \gamma_{i,j}) \log \left( \frac{1 - \alpha}{1 - \gamma_{i,j}} \right) + \gamma_{i,j} \log \left( \frac{\alpha}{\gamma_{i,j}} \right)}_{\text{Negative Spike KL Divergence}} \right]. \qquad (23)$$

This prior term presents two components. The first is the negative KL divergence between the distributions of the Slab variables, multiplied by the probability of $z_{i,j}$ being non-zero $\gamma_{i,j}$. The second term is the negative KL divergence between the distributions of the Spike variables. We find of particular interest that by computing the KL divergence analytically we recover a linear combination of the Spike and Slab components divergences.

## C  SPIKE AND SLAB DRAWS REPARAMETRISATION

### C.1  REPARAMETRISATION OF THE DRAWS

The draws $z_{i,l}$ are computed as follows

$$z_{i,l} = T(\eta_l - 1 + \gamma_i) \odot (\mu_{z,i} + \sigma_{z,i} \odot \epsilon_l), \qquad (24)$$

where $\odot$ indicates an element wise product. The function $T(y_{i,l})$ is in principle a step function centered at zero, however, in order to maintain differentiability, we employ a scaled Sigmoid function $T(y) = S(cy)$. In the limit $c \to \infty$, $S(cy)$ tends to the true binary mapping. In practice, the value of $c$ needs to be small enough to provide stability of the gradient ascent. In our implementation we employ a warm-up strategy to gradually increase the value of $c$ during training.

### C.2  SPIKE VARIABLE REPARAMETRISATION

We report here a detailed description of the Spike variable reparametrisation, similar to the relaxation of discrete variables in Maddison et al. (2016) and Rolfe (2016). Our aim is to find a function $f(\eta_{l,j}, \gamma_{i,j})$ such that a binary variable $w_{i,l,j} \sim p(w_{i,l,j})$ drawn from the discrete distribution $p(w_{i,l,j} = 1) = \gamma_{i,j}, p(w_{i,l,j} = 0) = (1 - \gamma_{i,j})$ can be expressed as $w_{i,l,j} = f(\eta_{l,j}, \gamma_{l,j})$, where $\eta_{l,j}$ is some noise variable drawn from a distribution which does not depend on $\gamma_{i,j}$.

The function of choice $f(\eta_{l,j}, \gamma_{i,j})$ should ideally only take values 1 and 0, as these are the only values of $w_{i,l,j}$ permitted by $p(w_{i,l,j})$. Furthermore, the probabilities of $w_{i,l,j}$ being 1 or 0 are linear in $\gamma_{i,j}$, therefore the distribution of the noise variable $\eta_{i,j}$ should have evenly distributed mass. The simplest function which satisfy these conditions and yields our reparametrisation is then

a step function $f(\eta_{l,j}, \gamma_{i,j}) = T(\eta_{l,j} - p(w_{i,l,j} = 0)) = T(\eta_{l,j} - 1 + \gamma_{i,j})$ where $\eta_{l,j}$ is uniformly distributed and $T(y)$ is the following step function

$$T(y) = \begin{cases} 1, & \text{if } y \geq 0. \\ 0, & \text{if } y < 0. \end{cases} \tag{25}$$

An illustration of this reparametrisation is shown in figure 6.

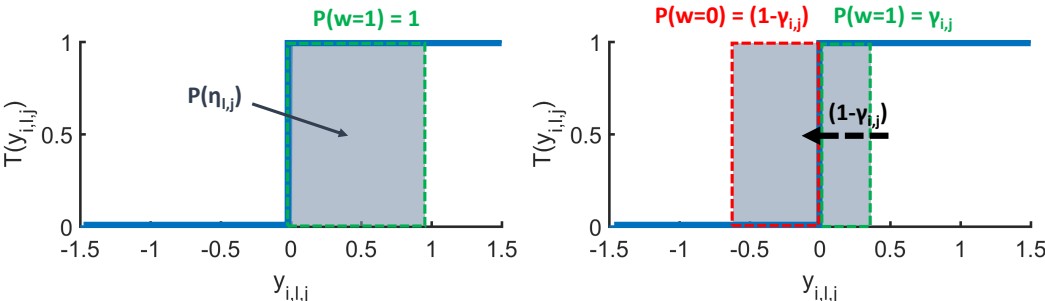

Figure 6: Schematic representation of the reparametrisation of the Spike variable. The variable $y_{i,l,j}$ is drawn in the range covered by the grey square with probability proportional to its height. On the left, for a spike probability $\gamma_{i,j} = 1$, the variable $y_{i,l,j}$ is drawn to always be greater than zero and the Spike variable $w_{i,l,j}$ is always one. On the right, for an arbitrary $\gamma_{i,j}$, the probability density of $y_{i,l,j}$ is displaced to the left by $1 - \gamma_{i,j}$ and $y_{i,l,j}$ has probability $\gamma_{i,j}$ of being $\geq 0$, in which case $w_{i,l,j}$ is one, and probability $1 - \gamma_{i,j}$ of being $< 0$, in which case $w_{i,l,j}$ is zero.

As described in section 3.2, the function $T(y_{i,l,j})$ is not differentiable, therefore we approximate it with a scaled Sigmoid $S(cy_{i,l,j})$, where $c$ is a real positive constant. In our implementation, we gradually increase $c$ from 50 to 200 during training to achieve good approximations without making convergence unstable.

## D DETAILS OF THE EXPERIMENTS

### D.1 DETAILS OF THE AUTO-ENCODERS

In our experiments, we use VAEs and VSCs having one 400-dimensional hidden layer between the observations and latent variables, both for encoders $q_\phi(z|x)$ and decoders $p_\theta(x|z)$. The only exception is the VSC used to obtain the qualitative results with the CelebA dataset, which was composed of two hidden layers with $2,000$ dimensions between the observations and latent variables.

We trained all auto-encoders with the ADAM optimiser, where the initial training rate was chosen according to best VLB performance of the standard VAE and kept the same for the corresponding VSC we compare to it. All training rates used were between 0.001 and 0.01.

### D.2 DETAILS OF THE DATASETS

MNIST and Fashion-MNIST are composed of $28 \times 28$ grey-scale images of hand-written digits and pieces of clothing respectively. Both sets contain ten different classes, which is the categories in which we classify in section 4.2. CelebA is a dataset of $200,000$ examples of colour images of celebrity faces.

We normalise the MNIST and Fashion-MNIST examples to have unitary norm before performing our experiments. For the CelebA examples, we use the centered version of the dataset, crop and downsample the images to obtain $32 \times 32$ RGB pictures, which we also normalise to have unitary norm before performing experiments.

We divide the datasets in training and test sets. For the MNIST and Fashion-MNIST sets, we preserve the original division of $60,000$ training examples and $10,000$ test examples. For the CelebA

dataset, we use a subset of $100,000$ examples as training set and one of $20,000$ as test set. For all results presented, the models were trained using the training sets and the results presented are obtained by encoding/decoding examples from the test sets, unless otherwise stated.

# E    ADDITIONAL EXPERIMENTAL RESULTS

## E.1    EXAMPLES OF SPARSE CODES AND RECONSTRUCTIONS

Figure 7 shows some examples of latent codes and reconstruction recovered with the VSC model at different values of prior sparsity $\alpha$.

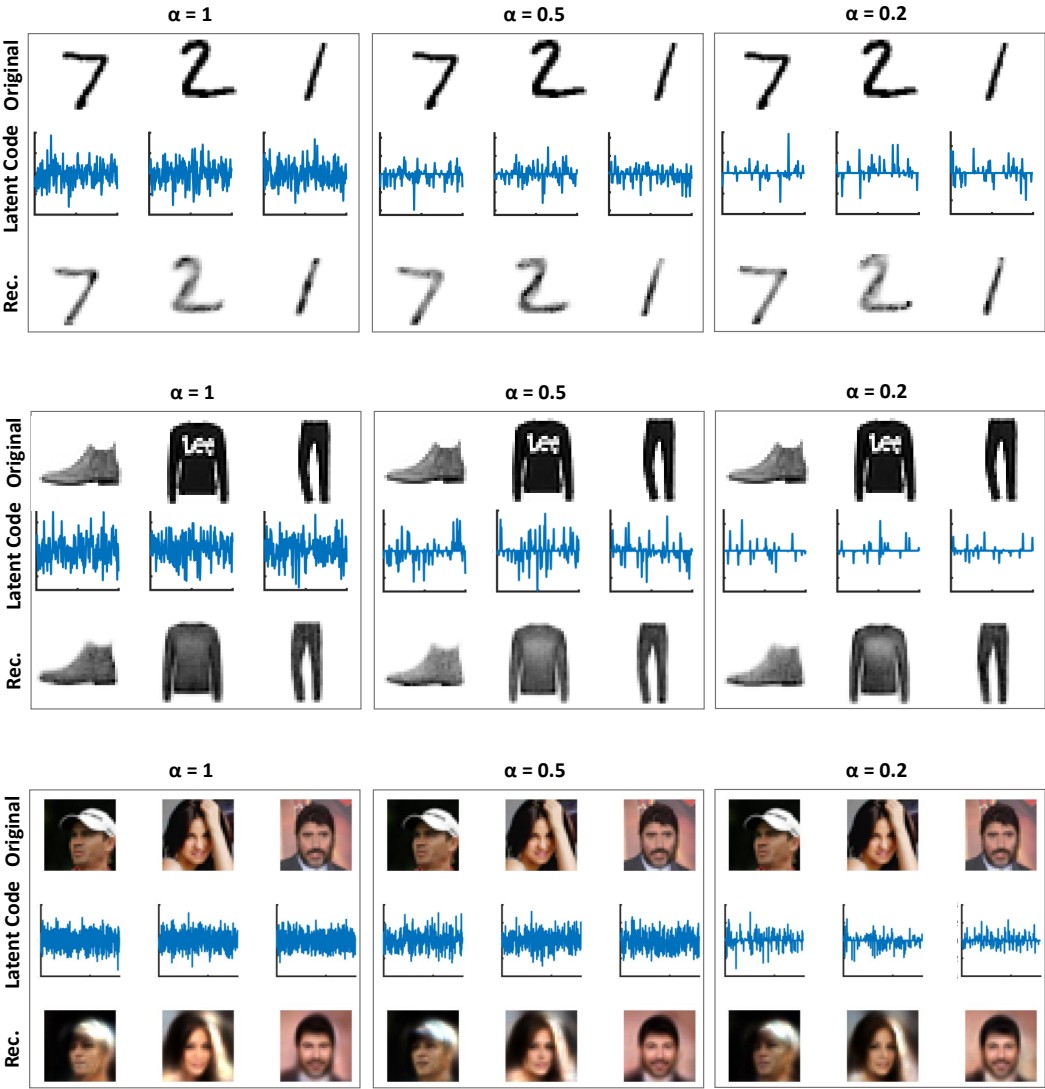

Figure 7: Examples of sparse codes and reconstruction for the MNIST (top), Fashion-MNIST (middle) and CelebA (bottom) datasets.

## E.2    LATENT SPACE SPARSITY

We measure the latent space posterior sparsity at varying prior sparsity $\alpha$. We encode both the MNIST and Fashion-MNIST datasets in 200-dimensional spaces with different values of the prior

Spike probability $\alpha$. In each case, we measure the aggregate posterior sparsity. Results are shown in figure 8.

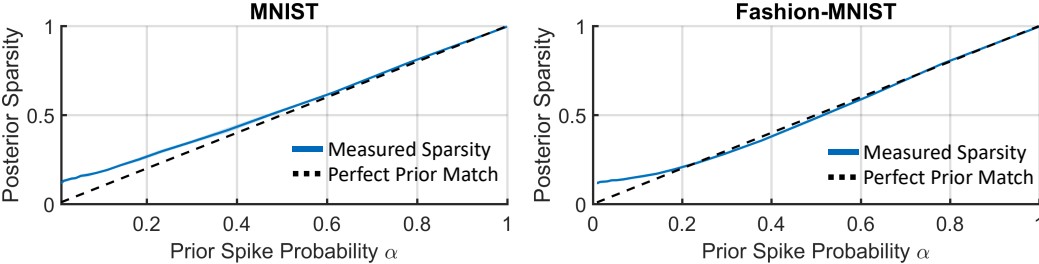

Figure 8: Measured sparsity at varying prior Spike probability $\alpha$.

For larger values of $\alpha$ the resulting codes retain approximately the sparsity induced by the prior as expected. At lower values of $\alpha$ the latent codes sparsity increasingly departs from the value induced by the prior. This is expected since below a certain sparsity value, the recognition function is induced to activate a certain number of latent dimensions in order to satisfy reconstruction.

### E.3 ADDITIONAL ELBO EVALUATION

We report on the ELBO evaluation results. First, we show analogous results to those in figure 2 for different imposed iterations limits in figure 9

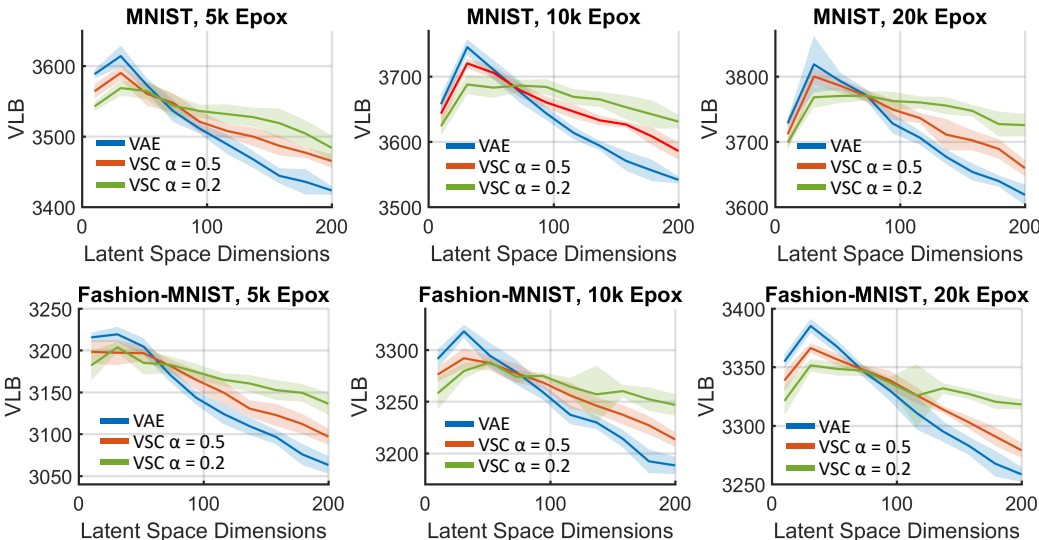

Figure 9: Test sets ELBO evaluation of VSC at varying number of latent space dimensions for different iterations limits. The out most right graphs correspond to those shown in figure 2.

Next, we show the behaviour of the lower bound at varying prior sparsity $\alpha$ for high dimensional latent spaces. We encode both the MNIST and Fashion-MNIST datasets in 200-dimensional spaces with different values of $\alpha$ and measure the training and test sets ELBO in each case. The results are shown in figure 10.

By making the Prior increasingly sparser (i.e. $\alpha$ going from 1 to 0) the ELBO increases thanks to the smaller sub-spaces needed to represent each observation. At very low $\alpha$, the lower bound decreases again, as the number of dimensions activated by the recognition function in order to describe the observations is too high to match the prior.

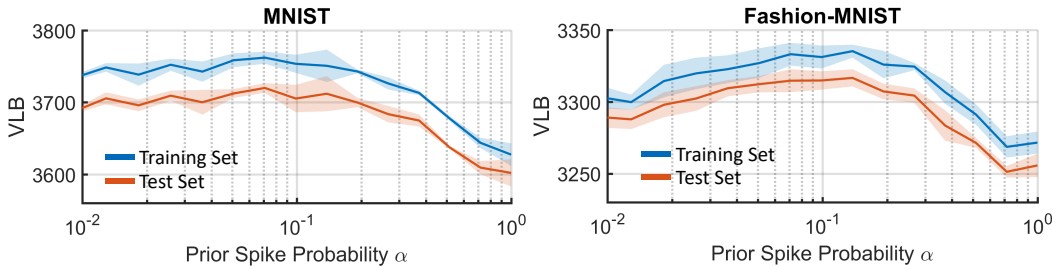

Figure 10: Training and test sets ELBO at varying prior Spike probability $\alpha$.

### E.4 CLASSIFICATION AT VARYING SPARSITY

We show the classification performance at varying prior sparsity $\alpha$ for high dimensional latent spaces and various limits of available number of training examples. We encode both the MNIST and Fashion-MNIST datasets in 200-dimensional spaces with different values of $\alpha$ and measure the classification accuracy when classifying with a one layer network as described in 4.2. Figure 11 displays the results.

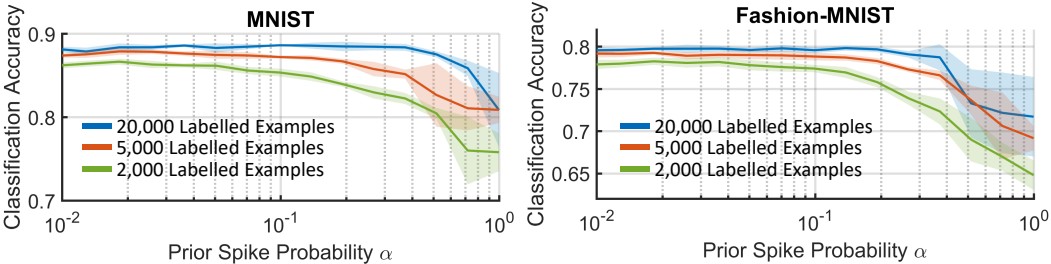

Figure 11: Classification performance at varying prior Spike probability $\alpha$.

As the prior Spike probability is decreased, the recovered codes are increasingly more efficient.

### E.5 ANCESTRAL AND CONDITIONAL SAMPLING

VAEs are attractive for their ability to produce arguably realistic samples from the prior through ancestral sampling. Though VSC can be used to perform the same generation task, samples directly from the sparse prior do not give as realistic synthetic observations, as not just any combination of sparse features is a feasible one (see figure 12).

However, VSC is capable of generating good synthetic samples conditioned on the combination of features identified in a particular observation. The recognition function from a certain observation defines a sub-space over certain active dimensions. If we sample from the Gaussian prior only along these dimensions we can generate objects that express variability only in the features recognised in the original observation. Examples are shown in figure 13.

**VAE Ancestral Sampling**          **VSC Ancestral Sampling (α = 0.1)**

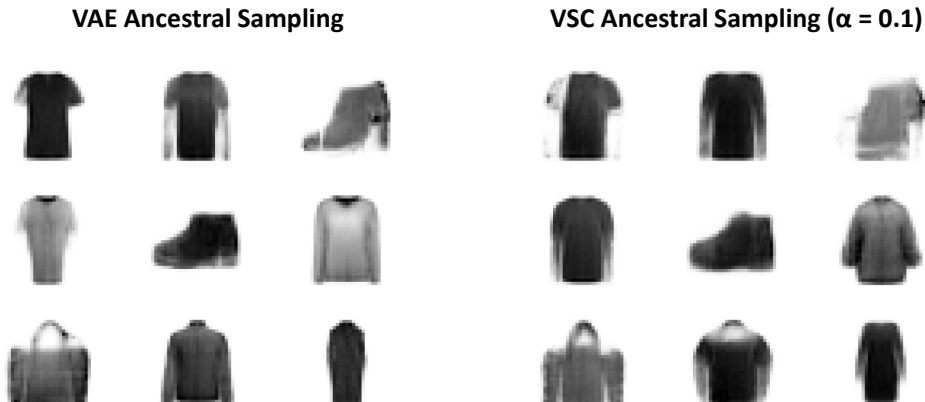

Figure 12: Ancestral sampling with a VAE and a VSC trained on the Fashion-MNIST dataset. The samples generated by the VSC sometimes result into unnatural combinations of features, such as asymmetric t-shirts and bags handles on pieces of clothing.

**VSC Conditional Sampling (α = 0.1)**

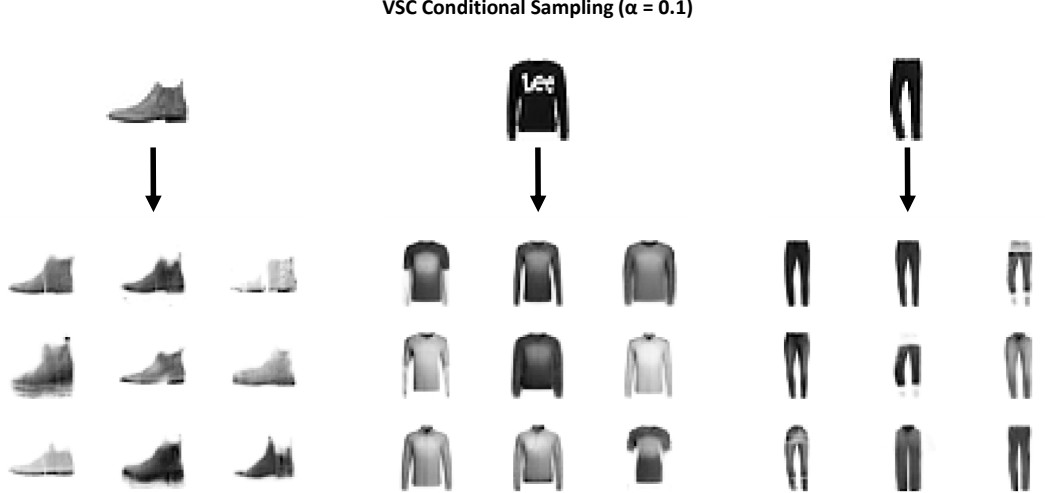

Figure 13: Conditional sampling in a VSC trained on the Fashion-MNIST dataset. Samples are drawn from the prior Gaussian component, but only along the dimensions activated by the recognition function, which are different for different observations. As a result, we obtain different sub-generative models that can generate different distinct types of objects in the aggregate used to train the model.

