# OpenReview forum: "Variational Sparse Coding"
_ICLR.cc/2019/Conference_

### Official Review · AnonReviewer3 · 2018-10-30
**Interpretable VAE with sparse coding**

**Rating:** 5
**Confidence:** 4

**Review:**

This paper presents variational sparse coding (VSC). VSC combines variational autoencoder (VAE) with sparse coding by putting a sparse-inducing prior -- the spike and slap prior -- on the latent code z. In doing so, VSC is capable of producing sparse latent code, utilizing the latent representation more efficiently regardless of the total dimension of the latent code, meanwhile offers better interpretability. To perform traceable inference, a recognition model with the same mixture structure as the spike and slap prior is used to produce the approximate posterior. Experimental results on both MNIST and Fashion-MNIST show that even though VSC performs comparably worse than VAE in terms of ELBO, the representation it learns is more robust in terms of the total latent dimension in a downstream classification task. Additionally, the authors show that VSC provides better interpretability by interpolating the latent code and find that some dimensions correspond to certain characteristics of the data.

Overall, the paper is clearly written and easy to follow. VSC is reasonably motivated and the idea behind it is quite straightforward. Technical-wise, the paper is relatively incremental -- all of the building blocks for performing tractable inference are standard: Since the posterior is intractable for nonlinear sparse coding, a recognition network is used; the prior is spike and slap, thus the recognition network will output parameters in a similar mixture structure with both a spike and a slap component; to apply reparametrization trick on the non-differentiable latent code, a continuous relaxation, similar to the one used in concrete distribution/Gamble trick, is applied to approximate the step selection function with a controllable "temperature" parameter. Overall, the novelty is not the strong suit of the paper. I do like the idea of VSC and its ability to learn interpretable latent features for complex non-linear models though. I have two major comments regarding the execution of the experiment that I hope the authors could address:

1. It is understandable that VSC is not able to achieve the same level of ELBO with VAE, as is quite common in models which trade off performance with interpretability.  However, one attractive property of VAE is its ability to produce relatively realistic samples right from the prior, since its latent space is fairly smooth. It is not clear to me if VSC has the same property -- my guess is probably not, judging from the interpolation results currently presented in the paper. It would be interesting if the authors could comment on this and maybe include some examples to illustrate it.

2. As is known in some recent literature, e.g. Alemi et al. Fixing a broken ELBO (2018), VAE can be easily trained to simply ignore the latent representation, hence produce terrible performance on a downstream classification task. I don't know exactly how the data is processed, but on MNIST, an accuracy of less than 90% means it is quite bad (I can get >90% with PCA + logistic regression). I wonder if the authors have explored the idea of learning better representation by including a scalar in front of the KL term -- or if VSC is more robust to this problem of ignoring latent code.

Minor comments:

A potential relevant reference: Ainsworth et al., Interpretable VAEs for nonlinear group factor analysis (2018).

---

> ### Author Response · Authors · 2018-11-14
> **Reply to Reviewer 3**
>
> 1 – *Sampling from the Prior* Sampling straight from the prior is not expected to produce as good quality samples as a standard VAE, since data samples are not well represented by just any combination of sparse features. However, samples from a PDF which has the Spike distribution of a conditional posterior (encoding from one observation) and slab distribution of the prior, not only produces good synthetic samples, but it does so conditioned on the features present in the particular encoded observation. Through the conditional activation of only certain variables, VSC defines a sort of “sub-generative model” for a given observation that models only the continuous sources of variations identified in the specific object and similar ones. For example, consider a VSC trained on fashion-MNIST; if we sample from the prior Gaussian, but only along the dimensions activated by the encoding of a t-shirt, we have a sub-generative model for t-shirts. We partially discuss this in section 4.3, however approaching it from a modification of encodings prospective rather than a sampling one. We realise the connection is not clear and we will add a discussion and experimental results either in the main body or appendix to clarify this important aspect.
>
> 2 – *Increasing the KL* Indeed representations found with VAEs do suffer from this known problem of ignoring latent representations. VSC does in part counteract this effect due to the discretisation from the spike variables, but is similarly affected by it. In our experiments we do not aim to obtain the best representations or classification accuracy achievable with our model, but rather compare to the standard VAE in order to highlight the difference between sparse and normally regularised latent vectors and the advantage in robustness when increasing the number of latent dimensions. The overall representation quality, and consequentially classification performance, can be improved at the expense of the ELBO value by increasing the coefficient of KL regularisation, as in beta-VAEs. By doing so, we get classification accuracies for MNIST above 90% for 5,000 labelled examples. We will add an experimental section in either the main body or the appendix where we compare this beta-VAE strategy for VAEs and VSCs and discuss how the VSC advantage varies as the beta coefficient is changed.

---

### Official Review · AnonReviewer2 · 2018-11-05
**Straightforward extension of VAEs to sparse priors**

**Rating:** 5
**Confidence:** 5

**Review:**

This paper proposes an extension of VAEs with sparse priors and posteriors to learn sparse interpretable representations. Training is made tractable by computing the analytic KL for spike and slab distributions, and using a continuous relaxation for the spike variable. The technique is evaluated on MNIST, Fashion MNIST, and CelebA where it learns sparse representations with reasonable log-likelihood compared to Gaussian priors/posteriors, but improved classification accuracy and interpretability of the representation.

While this paper is clear and well written, the novelty of the approach is limited. In particular, this is a straightforward application of vanilla VAEs with a different prior/posterior. The authors missed a bunch of related work and their main theoretical contributions are known in the literature (KL for spike and slab distributions, effective continuous relaxations for Bernoulli variables). The experiments are interesting but the authors should compare to more baselines with alternative priors (e.g. stick breaking VAEs, VampPrior, epitomic VAEs, discrete VAEs).

Strengths
+ Well written, clear, and self-contained paper. Figures are nice and polished.
+ Thorough experiments studying the effect of sparsity on the representation

Weaknesses
- No discussion/comparison to other VAE approaches that incorporate sparsity into the latents: Eptimoic VAEs (2017), discrete VAEs with binary or categorical latents are sparse (see: Discrete VAEs, Concrete/Gumbel-Softmax, VQ-VAE, output-interpretable VAEs), stick breaking VAEs, structured VAEs for the Beta-Bernoulli process (Singh, Ling, et al., 2017). Missing citation to foundational work on sparse coding from Olshausen and Field (1996).
- Lack of novelty: The analytic KL term for spike and slab priors has been derived before in Discrete VAEs (Rolfe, 2017) and in work on weight uncertainty (Yarin Gal's thesis, Blundell et al. 2016). Continous relaxations like the one used for the spike variable has been presented in earlier work (Concrete distributon, Gumbel-Softmax, Discrete VAEs).

Minor comments:
- Eq. 1, shape for B should be MxJ
- Cite Rezende & Mohamed for VAEs along w/ Kingma & Welling
- Definition of VAE is overly-restrictive. Typically a VAE is the combo of variational inference with an amortized inference network (and optionally reparameterization gradients). Saying that VAE implies Gaussian prior and Gaussian posterior is far too restrictive.
- VLB is a non-standard acronym, use ELBO for evidence lower bound
- I'm surprised that VAEs perform so poorly as latent dim increases. I'd expect it to just prune latent dimensions. Do you have an explanation for why performance drops for VAEs? Are they overfitting?
- VAEs with Gaussian p(x|z) are typically harder to train and more sensitive to hyperparameters than Bernoulli p(x|z). Could you repeat your experiments using the more common binarized MNIST so that numbers are comparable to prior work?
- If the goal is to learn representations with high information, then beta-VAEs or InfoVAEs should be compared (see analysis in Alemi et al., 2017). The number of dimensions may matter less for classification than the rate of the VAE. To analyze this further, you could plot the rate (KL(q(z|x) || p(z)) vs. the classification accuracy for all your models.
- Fig 4: consider adding in plots of continuous interpolation of the latent dimension (as in beta-VAE, TC-VAE, etc.)
- Would be interested to see how much class information is stored in the value vs. the pattern of non-zeroes in the latent representation (as done in Understanding Locally Competitive networks from Srivasta et al. 2014).
- Not at all expected as this came out after your submission, but would be nice to compare to a similar recent paper: https://www.biorxiv.org/content/early/2018/08/23/399246

---

> ### Author Response · Authors · 2018-11-14
> **Reply to Reviewer 2**
>
> *Contextualisation* We thank the reviewer for pointing out the related work. We agree that a better contextualisation is needed to appreciate the contribution. We will therefore modify the introduction and related work sections to incorporate relevant papers to the existing VAEs methods and different latent space priors.
>
> *Novelty* We do not claim novelty of the re-parametrisation trick for binary variables alone and we will cite the appropriate work as advised. However, we are unable to find in the literature referenced by reviewer 2 (and in general) the derivation of an analytic form for the general discrete mixture-Spike and Slab KL divergence (reported in section 3.1 and derived in appendix B of our paper).
>
> As we may be missing the relevant sections of the cited literature, we kindly ask reviewer 2 if he/she could refer to the specific pages or equations that detail an analytic form for the discrete mixture-Spike and Slab KL divergence we present in our paper ?
>
> In the mentioned works, we observe the following:
>
> -	In “Discrete Variational Auto-encoders” by Rolfe, the KL divergence term of the ELBO for a recognition function that models dependences between continuous and discrete variables is estimated and derived stochastically as detailed in appendix F. In our work, we derive directly an exact analytic discrete mixture-Spike and Slab KL divergence that induces sparse regularisation which does not require stochastic sampling to be estimated.
>
> -	In Yarin Gal’s thesis, the approximate posterior distribution q is the product of an approximation to an optimal posterior component obtained by moment matching and the prior itself (see p.124). The KL divergence between such approximate posterior and a Spike and Slab prior is then reported in appendix C. Because the approximate posterior q contains the prior p, this KL divergence is different and arguably simpler to compute analytically than the one we present in our paper; the prior simplifies inside the logarithm leaving the cross entropy between the approximate posterior (which contains a Spike and Slab) and the moment matched Gaussian (see p.159). In our work we derive a general discrete mixture-Spike and Slab KL divergence that works for any discrete-continuous mixture distribution recognition function.
>
> -	In “Weighted Uncertainty in Neural Networks” by Blundell et al. (if this is the paper reviewer 2 is referring to) the proposed prior is a scale mixture of two gaussians which resembles the Spike and Slab distribution (section 3.3) and the KL divergence is computed stochastically along with the rest of the ELBO (equation 2). While in this work the KL divergence is estimated by sampling from a general posterior q, we derive an exact analytic form for the KL divergence between a discrete mixture recognition function and a Spike and Slab prior.
>
> *Comparison with other VAE models* Experimental comparison with other priors presented in previous work would indeed be interesting. However, we point out that in our evaluation we aim to study the effect of sparsity in the latent space of a VAEs and show the characteristics of sparse representations rather than demonstrate a new method that performs better than previous ones in some settings. The comparison is drawn with respect to the standard VAE to clearly show how sparse latent representations differ from normally regularised ones and give the reader a clear intuition of what effects may be expected when inducing sparsity in the latent space and where it might be useful to do so in other models.

---

> > ### Author Response · Authors · 2018-11-14
> > **Reply to Reviewer 2 Minor Comments**
> >
> > We hereafter address what we believe to be the most relevant minor comments:
> >
> > -	We thank the reviewer for the corrections on the definitions and references in sparse coding and VAEs. We will revise the relevant sections in the paper accordingly.
> > -	Given a very large number of iterations and very small step size the standard VAE does approximately prunes extra dimensions when trained with a large latent space, still with some limited overfitting. However, within a limited iteration budget (of 20,000 iterations in our example) larger latent spaces fail to converge to a high enough value of the ELBO. The reason for the drop in classification performance is similar; with unlimited computational budget and many labelled examples the performance of VAEs is expected to only increase or stay stable as the latent space dimensionality increases, but with limited iterations and available labels overfitting and difficulty of convergence largely hinder performance.
> > -	We feel that comparing to the beta-VAE or info-VAE may be interesting, but we do not aim to compare the representation performance with these methods as they explore the theme of interpretation in a perpendicular direction. For instance, it is perfectly plausible to build a beta-VSC by varying the sparse prior term as it is done with other priors in the beta-VAE.
> > -	The paper “Sparse Coding Variational Auto-Encoders” is indeed related to our work and we thank the reviewer for pointing it out. The scope is however different; a variational autoencoder approach is used to obtain better inference in linear sparse coding and using heavy-tailed PDFs as sparsity promoting priors. In our work, we aim at modelling sparse non-linear features of observations with the Spike and Slab prior.

---

### Official Review · AnonReviewer1 · 2018-11-05
**Potentially Interesting**

**Rating:** 4
**Confidence:** 4

**Review:**

in this work the authors propose to replace Gaussian distribution over latent variables in standard variational autoencoders (VAEs) with a sparsity inducing spike-and-slab distribution. While taking the slab to be Gaussian, the authors approximate the spike with a scaled sigmoid function, which is then reparameterized through a uniform random variable. The authors derive an extension of the VAE lower bound to accommodate KL penalty terms associated with spikes. The variational lower bound (VLB) is optimized stochastically using SGD (with KL-divergence computed in closed form). Results on benchmarks show that as compared to standard VAE, the proposed method achieves better VLB for higher number of latent dimensions. Classification results on latent embeddings show that the proposed method achieves stable classification accuracy with increasing number of latent dimensions. Lastly the authors visualize sampled data to hint that different latent dimensions may encode interpretable properties of input data.

Originality and significance: In my opinion, the approach taken in this work does not constitute a major methodological advancement; the VLB authors derive is a relatively straight-forward extension of VAE's lower bound.

Pros:
The paper is well-written and easy to follow.
The idea of having a sparse prior in latent space is indeed relevant,
The approximation and reparameterization of the spike variable is however functionally appealing.
Potentially useful for semi-supervised learning or conditional generative modeling.

Concerns:
The authors show various empirical results to highlight the performance of their approach, but I am still not sure where it is best to use sparse embeddings that are induced by the proposed approach vs. those of standard VAE (or other of its sparse variants e.g., rectified Gaussian priors by Tim Salimans).  For instance in all experiments VAE seems to be competitive or better for low-dimensional latent space, so one may ask, why is it necessary to go to a higher number of latent variables? In a VAE setup, one can simply tune the number of latent dimensions through cross-validation, as one would probably need to do to tune the prior sparsity parameter in the proposed method.

I am also wondering if the disparity between VAE and proposed method w.r.t. classification performance for increasing number of latent dimensions vanishes as more labeled data is used for training? Fig. 11 in appendix seems to indicate that.

Lastly I am not sure how we can expect to always converge to interpretable encodings since there is nothing explicit in the objective function to encourage interpretable solutions. Perhaps samples such as those shown in the paper can also be generated by modulating VAE embeddings?

Maybe the proposed approach offers potential for tasks such as semi-supervised learning or conditional generative modeling, but the current set of empirical results does not allow one to draw any conclusions there.

---

> ### Author Response · Authors · 2018-11-14
> **Reply to Reviewer 1**
>
> We thank the reviewer for the comments and we are glad to find that some main points and advantages of our proposed method were recognised; inducing sparsity in the latent space of a VAE in order to find non-linear sparse codes that can constitute useful inputs in semi-supervised learning and allow for interpretable control in the generation of data. The novelty concern is addressed in the general reply, while below we address each individual concern:
>
> *Need for Cross-Validation* When finding useful latent representations for controlled generation and classification tasks there is no need to cross validate the sparsity parameter of the prior. In our experiments we set this parameter to a sufficiently low value (0.01) such that the regularisation term of the ELBO essentially induces the latent variables to be always zero and the reconstruction term induces only the variables it needs to reconstruct samples to be active. This effect occurs for any sufficiently low value of the prior sparsity parameter. This is shown in Fig.11 in the appendix, where for values of alpha lower than 0.1 the classification accuracy is steadily high.
>
> *Advantage with More Labels* The advantage is more pronounced at lower number of available labels and is especially useful in semi-supervised settings. However, the advantage is still present at higher regimes of labelled data; In figure 11 the blue line is the classification performance as a function of latent prior sparsity alpha for 20,000 labelled examples (1/3 of the examples used to train the VSC). At alpha=1, approximately corresponding to a standard VAE, the classification accuracy is ~81% for MNIST and ~72% for Fashion-MNIST, while it is ~88% and ~80% at alpha<0.1. We will clarify this point in the revised version of the paper.
>
> *Interpretation* The discussion on this aspect of sparse latent spaces is particularly interesting and we hope to initiate a conversation on it as well as study it formally in future work.
> We do not explicitly induce interpretation. However, sparsity in the latent space does result into a higher expectation of interpretability in large latent spaces, provided that the sources of variations in the observed data can be considered sparse (many possible features are present in the ensemble but only small subsets of them are present in each individual example).
> Consider a VAE with a large dimensionality of latent space. The model will cluster distinct objects in different regions of the latent space and controlled generation is possible by interpolating between the regions of an aggregate posterior. However, given the encoding for one single example, the direction in which to move to modify interpretable aspects of the generation is difficult to find; there are many normally distributed latent variables and interpretable changes may or may not be caused by altering any combination of these. Of course, it is possible to improve the expected interpretation of altering elements by lowering the dimensionality of the latent space, but this also reduces the capacity of the model and hiders the ability of modelling data that may present a large number of features in its aggregate.
> VSC aims at modelling data which presents few features in individual examples, but many in the data aggregate. When encoding a single example the vector we obtain only has a small subset of active features and we can expect these few dimensions to control the continuous variables that represent relevant sources of variation for this example and similar objects, while ignoring others by setting them to zero. In such a way the sub-space of smoothly variable features relevant to each encoded example is defined by the encoding itself. At the same time, the model retains the capacity to describe complicated data ensembles by being able to use different sparse elements for different examples.
> We realise this may not be very clear in the current version of the paper and we will make such theme a central point in the discussion of section 4.3.

---

### Public Comment · (anonymous) · 2018-10-05
**Should benchmark against prior work**

Previous work on variational inference for spike and slab sparse coding was evaluated on datasets such as CIFAR-10 and STL-10: datasets consisting of color photographs. That was 6+ years ago, when GPUs were significantly slower. If the proposed method actually works and scales, it should be possible to easily outperform papers from 2012 using modern hardware on the same benchmarks they used.

---

> ### Public Comment · (anonymous) · 2018-10-10
> **Citation**
>
> It would be very helpful to know, which works the previous commenter is exactly referring to. After all, the original paper on the VAE by Kingma and Welling was published only in 2013.

---

> ### Author Response · Authors · 2018-10-11
> **Difference in Scope**
>
> If you are referring to the paper from 2012 titled “Large-Scale Feature Learning With Spike-and-Slab Sparse Coding” we would like to point out that our work is quite different in scope. In this work from 2012, the authors propose an efficient Spike and Slab variational inference method for linear sparse coding models; a Bayesian parallel to traditional sparse coding if you will. The aim there is to induce regularisation in the recovery of sparse codes, in turn improving the reliability of feature extraction in images when classifying with a low number of labelled examples available. To this end, in their classification evaluation, they divide the images in small patches (6x6 I believe) and use these to learn the dictionary of sparse features.
>
> The aim of our work is to perform sparse variational inference with arbitrarily complicated non-linear mappings. By modelling non-linear sparse features we aim to obtain interpretable and useful latent representations while at the same time retaining the reconstruction/synthesis capability of generative models, rather than just extracting features. To make this tractable, we use the framework of VAEs (introduced in 2013, as mentioned by the commenter below). The inference we perform is significantly more computationally difficult than the feature extraction presented in the aforementioned previous work, mainly for two reasons:
>
> 1)	The model we use is non-linear, using neural networks in the mappings between latent and observation spaces, making variational inference way less tractable (hence the VAE approximate inference architecture).
>
> 2)	We don’t make any image assumption about the objects we model and use the raw entire images (MNIST and fashion-MNIST are 28x28=784 and our CelebA dataset is 32x32x3=3072 as opposed to the 6x6x3=108 pre-processed colour patches modelled in the previous paper).
>
> The combination of these two aspects allows us to isolate few global and non-linear sparse features, such as facial traits and clothes characteristics, rather than large dictionaries of linear sparse features over image patches, containing lines and curves such as those shown in the paper from 2012.
>
> To adapt our model specifically for classification of varied natural images (such as CIFAR) and benchmark against the strategy employed in the linear Spike and Slab inference paper, one could use convolutional  encoding and decoding neural networks with pooling regions of appropriate size and some pre-processing. Though this may certainly be an interesting investigation, it is beyond the scope of the work we present here; in our evaluation we are interested in examining the effect of sparsity in the latent space on the performance of general non-linear representation models (in particular VAEs and we use fully connected layers for generality) and not specifically improve the accuracy or computational efficiency for image classification.

---

### Author Response · Authors · 2018-11-14
**General Response**

We thank the reviewers for their thoughtful comments. We notice that the reviewers are mainly concerned with the novelty of our approach and the resulting algorithms. Our response to this criticism is as follows:

*Novel and Non-Trivial Analytic Contribution*: The derived ELBO is elegant and the resulting implementation intuitive; however, it is rigorously derived in a way that is not at all obvious, trivial or known in the literature (to the best of our knowledge; see individual replies for details). We derive directly an analytic expression for a discrete mixture-Spike and Slab KL divergence (reported in section 3.1 and derived in appendix B) which results in closed form variational sparse inference in a continuous space that gives rise to a distinctly different and simpler algorithm than previous approaches. We will include a concise version of the KL derivation of appendix B in the main body of our revised paper.

*Intuitive and Generalisable Approach*: Formulating the problem in analogy to the original VAE is intentional for clarity and focus of scope. We present a general formulation of sparse inference with VAEs that can be a powerful tool to obtain sparse representations and is extendable in different directions. The presented ELBO can be incorporated in more elaborate models that aim to infer sparsity thus it is not a stand-alone model to solve a specific problem.

We will revise our paper to clarify these key points and relate the contribution more explicitly to previously works (e.g. outlined by R2) which have similar overall goals but approach the problem in distinctly difference ways.

We will post replies to each individual reviewer. We aim to present an updated version of our paper by this Friday, 16th November, but would invite the reviewers to comment on our feedback and current plans as soon as possible.

---

> ### Author Response · Authors · 2018-11-17
> **Revision**
>
> We have now uploaded a revised version of our paper. We have made the following changes:
>
> -	*Additional Related Work Section* We have added a subsection (2.3) covering related work on discrete VAEs and sparsity in VAEs.
> -	*KL Divergence Derivation* We have included a derivation of the analytic KL divergence term we present and use in our ELBO in section 3.1.
> -	*More detailed discussion on interpretation* We have extended the discussion in section 4.3 to make clearer the intuition behind the expected improved interpretation of VSC.
> -	*Supplementary on Sampling* We have added a supplementary section (E5) showing the difference in ancestral sampling between VAEs and VSC and the ability of VSC to perform conditional sampling.
>
> We will apply some further modifications in response to other remarks, but would invite the reviewers to comment on these main updates and the current version of the paper.

---

### Public Comment · ~Alfredo_De_la_Fuente1 · 2019-01-07
**Reproducibility study of Variational Sparse Coding paper**

As part of the ICLR 2019 Reproducibility Challenge, we worked to reproduce the results reported in this paper (Variational Sparse Coding). Given no available code for the project, we implemented the variational autoencoder architecture described in the paper from scratch.  We validate the experimental results and further propose improvements or future research directions that may contribute to the machine learning community. A link to the full report, as well as the repository with code, can be found at the end of this message.

The authors' main motivation for this work lies in developing a model able to learn sparse representations that are informative (for further classification tasks) and interpretable (by exploring the latent sparse space). In this line of thought, they propose an improvement over the Variational AutoEncoder model, explicitly modeling sparsity in the latent space with a Spike and Slab prior distribution and drawing ideas from sparse coding theory.

Overall, the paper describes in enough detail the Variational Sparse Coding (VSC) model implementation. Only some details in the optimization hyperparameters and initialization mechanisms would be ideal. However, given enough training epochs, the VSC model we implemented was able to converge and produce the desired results in the different tasks described in the paper.

Further development on testing the model and comparing it against other sparse models on well-known benchmarks is critical. In order to assess how interpretable the learned latent features are, we could extract ideas from disentangled representations, to measure the effect of sparsity on a disentanglement metric, against benchmark models such as β-VAE or Factor-VAE. Without a proper benchmark, we are not able to understand how the learned representations can be interpretable. Although for small latent dimensions, visual inspection is enough, there must be a metric for comparison.

Finally, we suggest using convolutional architectures to obtain less blurry images and richer sparse representations. In addition, by applying the model to the Disentanglement testing Sprites dataset, we may observe and measure the interpretability of the learned latent sparse representations.

We conclude that the model hypothesis was validated through corroborating results in different experimental setups with our model implementation. Therefore, the research paper is reproducible.

Full report : https://drive.google.com/open?id=1sEmiD2_dOwTJVydIsiZ1bsxdtbaBQT_Z
Code: https://github.com/Alfo5123/Variational-Sparse-Coding

---

> ### Author Response · Authors · 2019-02-17
> **A clear and complete reproducibility report**
>
> We thank the authors of the reproducibility study for their efforts to reproduce our results and test our findings. The report is thorough, clear and complete. We are glad to see that our main conclusions were confirmed by this study and appreciate the sensible suggestions, which we are very much taking into account to extend and improve our work.
>
> Regarding some of the minor missing information that were pointed out in the report:
> -	The batch size used in the experiment reported in the paper was 50 for the MNIST and Fashion-MNIST experiment and 100 for the CelebA experiments. We are happy to see that this choice was not critical, as the reproducibility study was carried out using batches of 32 samples instead.
> -	The multiplicative weights were initialised as normal random variables with variance 1, while additive weights for each layer were initialised as zeros.
> -	We thank the authors of the study for elaborating on the resources cost of the technique, which we did not investigate in our work. We carried out our experiments with a Titan X GPU and the running times seem to correspond to our experience.
>
> Regarding the suggestion made, these are all relevant and we agree with them to a large extent. More specifically:
> -	As suggested, In our future extension of the work we will ensure that every considered model stably reaches a local minimum.
> -	If we again demonstrate our methods solely with images, we will improve the decoding architectures to more specifically adapt to image statistics, using convolutional layers and conditional pixel structures, such as those used in PixelVAEs. Alternatively, we will broden the scope of our work to make it more general, testing also with other data, such as sound, speech, pose estimation or text.
> -	We will definitely test the method with a feature disentanglement benchmark dataset. We believe the main advantage of VSC to be the unsupervised discovery of features, therefore a quantitative evaluation of this capability is indeed very relevant.

---

### Meta-Review · Area_Chair1 · 2018-12-19
**Clean straight-forward derivation of VAE with spike-and-slab prior. Well-written but incremental and too limited in scope.**

**Confidence:** 4
**Recommendation:** Reject

**Metareview:**

The paper develops and investigates the use of a spike-and-slab prior and approximate posterior for a VAE. It uses a continuous relaxation for the discrete binary component in the reconstruction term of the ELBO, and an analytic expression for the KL term between the spike-and-slab prior and approximate posterior. Experiments on MNIS, Fashion-MNIST and CelebA convincingly show that the approach works to learn sparse representations with improved interpretability that also yield more robust classification

All reviewers agreed that this approach to sparsity in VAEs is well motivated and sound, that the paper is well written and clear, and the experiments interesting.
One reviewer noted that the accuracy on MNIST remains really poor, so the approach does not cure VAEs yielding subpar representations for classification (although not the goal of this research).

The reviewers and the AC however all judged that it currently constitutes a too limited contribution because a) the approach is a straightforward application of vanilla VAEs with a different prior/posterior, and is thus rather incremental. b) the scope of the paper is rather limited, in particular as it does not sufficiently discuss and does not empirically compare with other (VAE-related) approaches from the literature that were developed for sparse latent representations.